# Microbial Pathogens in Aquaponics Potentially Hazardous for Human Health

**DOI:** 10.3390/microorganisms11122824

**Published:** 2023-11-21

**Authors:** Toncho Dinev, Katya Velichkova, Antoniya Stoyanova, Ivaylo Sirakov

**Affiliations:** 1Department of Biological Sciences, Faculty of Agriculture, Trakia University, 6000 Stara Zagora, Bulgaria; genova@abv.bg; 2Department of Plant Production, Faculty of Agriculture, Trakia University, 6000 Stara Zagora, Bulgaria; toni_1219@abv.bg; 3Department of Animal Husbandry–Non-Ruminant Animals and Special Branches, Faculty of Agriculture, Trakia University, 6000 Stara Zagora, Bulgaria; ivailo_sir@abv.bg

**Keywords:** pathogenic bacteria, yeasts, molds, aquaponics, human health

## Abstract

The union of aquaculture and hydroponics is named aquaponics—a system where microorganisms, fish and plants coexist in a water environment. Bacteria are essential in processes which are fundamental for the functioning and equilibrium of aquaponic systems. Such processes are nitrification, extraction of various macro- and micronutrients from the feed leftovers and feces, etc. However, in aquaponics there are not only beneficial, but also potentially hazardous microorganisms of fish, human, and plant origin. It is important to establish the presence of human pathogens, their way of entering the aforementioned systems, and their control in order to assess the risk to human health when consuming plants and fish grown in aquaponics. Literature analysis shows that aquaponic bacteria and yeasts are mainly pathogenic to fish and humans but rarely to plants, while most of the molds are pathogenic to humans, plants, and fish. Since the various human pathogenic bacteria and fungi found in aquaponics enter the water when proper hygiene practices are not applied and followed, if these requirements are met, aquaponic systems are a good choice for growing healthy fish and plants safe for human consumption. However, many of the aquaponic pathogens are listed in the WHO list of drug-resistant bacteria for which new antibiotics are urgently needed, making disease control by antibiotics a real challenge. Because pathogen control by conventional physical methods, chemical methods, and antibiotic treatment is potentially harmful to humans, fish, plants, and beneficial microorganisms, a biological control with antagonistic microorganisms, phytotherapy, bacteriophage therapy, and nanomedicine are potential alternatives to these methods.

## 1. Introduction

The rapid increase in world population has led to several challenges, such as water scarcity, degradation of arable land, and climate change, which in turn affects food production. However, conventional agriculture is renowned for requiring a large amount of water and land and high nutrient intake. Crops are also more likely to be infected by soil-borne diseases as a result of cultivation that also degrades the quality of the soil. To bridge the mismatch between resource availability and food demand, a shift from an infinite growth-based economic model to balanced and sustainable food production methods to ensure food security is required [1,2].

The Food and Agriculture Organization of the United Nations (FAO) defines aquaculture as the cultivation of aquatic organisms in both fresh and salt water under controlled conditions. On the other hand, hydroponics is the cultivation of plants without the use of soil, in which the nutrients necessary for the growth of the plants are supplied through an aqueous solution. The union of aquaculture and hydroponics is a comparatively new technology named aquaponics [3]. An aquaponic system could be either coupled or decoupled. A coupled system is designed so that water returns to the fish tank following plant irrigation. In decoupled system the produce irrigation water does not return to the fish tank [4]. Compared to soil-based farms, hydroponic cropping systems are not restricted by climate or location, better utilize vertical spaces, save approximately 90% irrigation water, and can supply fruit and vegetables to surrounding communities year-round. In addition, hydroponically grown plants have a higher yield and amount of some bioactive compounds than conventionally grown ones. On the other hand, freshwater fish excrete nutrients in their gills, urine, and feces waste streams, many of which are essential for plant growth and development in aquaponic systems. Thus, hydroponics and aquaponics are viewed as a promising solution for feeding the growing global population [1,5,6]. In addition to coupled aquaponics, the recent trends showed an increasing focus on the development of inland saline and marine aquaponics as the future of this farming system [7].

Although aquaponics is an innovative system, aquaponic products are already commercially available in many countries such as the USA, the UK, Canada, Australia, Israel, Philippines, Italy, India, South Africa, Uganda, Hungary, Portugal, Oman, China, Brazil, Ghana, Malaysia, Malta, Mauritius, and Japan. The most commonly raised aquatic animals by percentage are tilapia (*Tilapia* spp.), ornamental fish, catfish (order Siluriformes), other aquatic animals (such as shrimp and prawns, suborder Dendrobranchiata, and crayfish, Astacoidea and Parastacoidea families), perch (*Perca* spp.), bluegill (*Lepomis macrochirus*) trout (*Oncorhynchus* spp., *Salmo* spp., *Salvenlius* spp.), bass (*Micropterus* spp., *Morone* spp.), carp (Cyprinidae family), and other fish. The most frequently raised plants among commercial producers are basil, salad greens, non-basil herbs, tomatoes, head lettuce, peppers, cucumber, and other plants [8].

Along with plants and fish, microorganisms are also present in aquaponics. Bacteria are key players in processes which are fundamental for the functioning and equilibrium of aquaponic systems. For example, the process of nitrification is carried out mainly by ammonia-oxidizing and nitrite-oxidizing bacteria. Microorganisms can also contribute to the extraction of various macro- and micronutrients from the feed leftovers and solid feces and make them available for plant intake [9]. Also, aquaponics can be similar to soil production in terms of microbial communities [10]. However, in aquaponic systems there are not only beneficial, but also potentially harmful microorganisms of fish, human, and plant origin. In this respect, a coupled aquaponic system is a potential risk factor because the water circulates in the system, thus providing a perfect environment for waste accumulation, leading to continuous microbial and parasitic growth, i.e., outbreak of diseases [6]. The removal of pathogens is difficult if no mitigation methods are established. These methods are restricted in aquaponic systems because if they are able to reduce pathogenic microbial load, they are likely to reduce beneficial microbial load as well. Nevertheless, potential mitigation methods that can help reduce the pathogenic microbial loads, such as UV irradiation, ozonation, and filtration, have been explored [4].

The term “zoonosis” refers to a disease that can be transferred from animals to humans. Human infections caused by pathogens transmitted from fish or the aquatic environment are quite common and depend on the season, the patient’s contact with fish, and the associated environment, dietary habits, and state of the immune system of the exposed individual [11]. Such zoonotic infections can be divided into two categories: first, foodborne infections caused by eating raw or undercooked fish and ingesting water or other substances contaminated with infected fish feces/mucus and second, topically acquired infections caused by contact with fish pathogens through open wounds or skin scratches/abrasions [12,13]. Each year in the United States approximately 260,000 people get sick from contaminated fish. In the period between 1998 and 2015, a review on the data of the Centers of Disease Control and Prevention of the United States found 857 foodborne disease outbreaks associated with fish, resulting in 4814 illnesses, 359 hospitalizations, and four deaths. Most hospitalizations were caused by *Salmonella* spp. (31%) and ciguatoxin (31%) [14].

Many plant pathogens exist, but until recently the ability of plant pathogens to cause disease in humans and animals was thought to be of little importance. However, recent evidence suggests that animal and human infections caused by plant pathogens (bacteria, viruses, and fungi) may have critical impacts on human and animal health and safety. In the majority of cases, such infections result from infection through damaged skin or the respiratory tract, or were infections of immunocompromised individuals. As such, infections by phytopathogens can generally be considered opportunistic [15]. In the last several decades, multiple foodborne disease outbreaks have been traced back to the pre-harvest period contamination of fresh plant produce. In the period between 1973 and 2012, the Centers for Disease Control and Prevention of the United States received report of 606 leafy vegetable-associated outbreaks, with 20003 associated illnesses, 1030 hospitalizations, and 19 deaths. The pathogens most commonly causing leafy vegetable-associated outbreaks were norovirus (55% of outbreaks with confirmed etiology), Shiga toxin-producing *Escherichia* coli (18%) and *Salmonella* spp. (11%) [16].

In view of the aforementioned facts, the aim of the present review is to list the microbial pathogens found so far in aquaponics and to analyze their ways of transmission and potential hazard to human health as well as the proper methods of their control.

## 2. Microbial Pathogens in Aquaponic Systems

### 2.1. Bacteria

Bacteria are considered the main zoonotic agents of fish [10]. However, farmed aquatic species are poikilothermic with a labile body temperature dependent on the environment temperature. Their body temperature is generally too low to be considered optimal for the proliferation of most intestinal bacteria likely to infect humans because they prefer the body temperature of homeotherms [17]. Nevertheless, the rearing of species such as tilapia *(Oreochromis niloticus)* could allow the proliferation of introduced human pathogens because of the warm temperatures (28–30 °C) required for the optimal growth of these species [18]. However, the recent literature data included a case of an increasing count of *E. coli*, coliforms, Enterobacteriaceae, and *Salmonella* spp. from the beginning of the trials to their end while the temperature of the water was 18 °C [19]. These findings suggest that the restrictive effect of water temperature on the proliferation of intestinal bacteria is not absolute. In this respect, Lee et al. [20] found that if *E. coli* is provided with adequate nutrients, it is still able to grow at temperature as low as 15 °C, but at a reduced rate. Also, in the last years there have been a lot of data suggesting the increasing role of *E. coli* and coliforms in aquaponic systems [4,19,21].

There are hundreds of types of non-fecal coliform bacteria in the air, water, and soil, as well as fecal coliform bacteria represented mostly by *E. coli* in the waste of all humans, mammals, and some birds. If present in aquaponic systems, indicator and pathogenic bacteria, such as *E. coli* and *Salmonella* spp., most likely originate from warm-blooded animals and humans, since these enteric bacteria are transient in fish gut microflora (Table 1) [22]. However, the vast majority of these coliforms are completely harmless [23]. Nevertheless, coliforms such as *E. coli* and *Klebsiella*, and members of *Salmonella* are known as zoonotic agents of fish. They usually cause topically acquired and/or systemic infections in humans through contact via open wounds, touching fish, or skin scratches, or through foodborne infections via consumption of *Salmonella*-contaminated fish [13]. Because of their high pathogenic potential, the presence of *Salmonella* spp. in aquaponic water is considered highly hazardous. It should be noted that the subspecies of *Salmonella* found in the environment and cold-blooded animals are rare in humans. The four different clinical manifestations of human salmonella infection are enteric fever, gastroenteritis, bacteremia and other extraintestinal complications, and a chronic carrier state [24]. According to the World Health Organization (WHO), *Salmonella* belongs to the list of high priority pathogens which lists pathogens for which new antibiotics are urgently needed. *Salmonella* is resistant to fluoroquinolones [25]. *Klebsiella* spp. are found ubiquitously in nature, including in plants, animals, and humans. They cause several types of infections in humans, including respiratory tract infections, urinary tract infections, and bloodstream infections [26]. *K. pneumoniae*, one of the most important species of this genus with pathogenic potential, is not only a highly dangerous opportunistic human pathogen but in the last decades many strains of this bacterium demonstrated multidrug resistance, extensive drug resistance, and pandrug resistance, making the treatment of *K. pneumoniae* infections a real challenge. According to the WHO, *Klebsiella* belongs to the most critical group of pathogens for which new antibiotics are urgently needed. *Klebsiella* is carbapenem-resistant and ESBL-producing [25,27,28]. Moreover, *K. pneumoniae* is a fish pathogen infecting various fish species, including Nile tilapia *(Oreochromis niloticus)*, ornamental Nishikigoi carp (*Cyprinus carpio*), and Indian major carp (*Labeo rohita*). [29,30].

It is established that poor worker hygiene practices can lead to the infiltration of various microorganisms, including pathogens, into aquaponic systems. For example, if the workers of these farms do not switch shoes when entering the farm, even if the sanitizer sink is presented at the entrance it may not eliminate the risk of introducing contaminants from the open environment [6]. The discovery of *E. coli* in these systems is not necessarily a human health hazard because most of the strains of this bacterium are harmless or opportunistic. This, however, is not the case with Shiga toxin-producing *E. coli* which are found by Wang et al. [21] in aquaponic water. In humans, this bacterium can cause bloody diarrhea, stomach cramps, and vomiting followed by a serious sequela, hemolytic uremic syndrome (HUS), a condition characterized by thrombocytopenia, hemolytic anemia, and kidney failure [31]. Moreover, although *E. coli* is intrinsically susceptible to almost all clinically relevant antimicrobial agents, this bacterium has a great capacity to accumulate resistance genes, mostly through horizontal gene transfer resulting in multidrug resistance. Therefore, in the WHO list, *E. coli* belongs to the first priority critical group of pathogens for which new antibiotics are urgently needed. *E. coli* is carbapenem-resistant and ESBL-producing [25,32]. The human health hazards are especially serious when considering the potential for internalization of *E. coli* (including *E. coli* O157:H7) and *Salmonella* spp. that has been demonstrated in lettuce, spinach, and tomatoes grown in an inoculated hydroponic system [33,34,35,36]. However, Macarisin et al. [36] found that experimental contamination of spinach plants grown in soil resulted in a greater number of internalization events than in those grown hydroponically, suggesting that *E. coli* O157:H7 internalization depends on root damage, which is more likely to occur when plants are grown in soil. This hypothesis is supported by the fact that injury to the root system in hydroponically grown spinach increased the frequency of *E. coli* O157:H7 internalization [36]. Generally, the pathogens could penetrate the plant through the stomata, the roots and the damaged skin surface. Thus, the fact that the edible part of the plant is not exposed to fish feces in an aquaponic system is of paramount importance. Even if an aquaponically grown plant containing internalized pathogens is taken for consumption, disease is unlikely to follow because the roots containing the pathogens are usually removed [22]. This is a very important food safety advantage of plants grown in aquaponics compared to those grown in soil.

Contrary to the aforementioned experiments, Wang et al. [21] did not find the presence of *Salmonella* spp. and *Listeria monocytogenes* in aquaponic water and the contaminated water did not lead to internalization of Shiga toxin-producing *E. coli* into the roots, leaves, and fruit of the lettuce, basil, and tomato plants studied. Also, while some authors reported the presence of *E. coli* and *Salmonella* in aquaponic systems [19,37], others did not find them [6,21]. Several investigations did not find one or more of the dangerous human pathogens *E. coli* O157:H7, *Salmonella* spp., and *L. monocytogenes* in aquaponic systems [6,22,38]. The abovementioned differences found in the various studies are probably due to the hygiene and the disease prevention practices applied in the different aquaponic systems.

*Plesiomonas* spp., the existence of which are found in aquaponics (Table 1), are recently re-categorized from the Vibrionaceae family to the Enterobacteriaceae family, in which it is the only oxidase-positive member. *Plesiomonas shigelloides* most often causes enteritis in humans while the other illnesses associated with this bacterium include septicemia and central nervous system disease, eye infections, and a variety of miscellaneous ailments. The highest number of cases of *P. shigelloides* enteritis are reported in Southeast Asia and Africa. This bacterium is highly pathogenic to farmed fish [39,40].

*Shigella sonnei*, an important member of Enterobacteriaceae, is sometimes associated with vegetable-related foodborne disease outbreaks. In susceptible hosts, *Shigella* causes dysentery with typical symptoms of diarrhea, abdominal pain, vomiting, and fever. In the majority of cases, infected food workers are considered the main source of *Shigella* [41]. According to the WHO, *Shigella* spp. belong to the list of medium priority pathogens for which new antibiotics are urgently needed. *Shigella* spp. are fluoroquinolone-resistant [24]. In addition, raw fish and oysters are among the most commonly implicated foods in the transmission and cases of *Shigella* infection [42].

*Aeromonas* spp., pathogenic for fish and humans, include *A. hydrophila, A. caviae, A. jandaei, A. sorbia, A. salmonidae,* and *A. veroni*; among them the most common pathogen is *A. hydrophila* [10]. According to Jin et al. [43] *Aeromonas hydrophila* virulence factors include enzymes, enterotoxin, adhesion, hemolysin, flagella, lipopolysaccharide, secretory systems, and quorum sensing. In humans, *Aeromonas* can cause gastrointestinal tract disorders and wound and soft tissue infections, as well as septicemia [44]. The multidrug-resistance of *Aeromonas* species is evidence of an emerging health problem in both humans and aquatic animals [45,46]. *Aeromonas* infections (including *A. hydrophila* infections) were the most widespread bacterial diseases occurring throughout the year in carps such as *Catla catla*, *Labeo rohita*, *Cirrhinus mrigala*, and *Hypophthalmichthys molitrix* [29].

The genus *Acinetobacter* includes a complex and heterogeneous group of bacteria, many of which are causing a range of opportunistic, often catheter-related, infections in humans [47]. The important member of this genus, *Acinetobacter baumanii*, is an emerging opportunistic pathogen in human medicine and a major cause of nosocomial infections worldwide. It is well-known for its ability to form biofilms, its strong environmental adaptability, and, especially, its multidrug resistance. According to the WHO, *A. baumannii* belongs to the most critical group of pathogens for which new antibiotics are urgently needed. *A. baumannii* is carbapenem-resistant. The most frequent clinical manifestations of *A. baumannii* infection in patients are pneumonia and bacteremia [25,48]. In addition, *A. baumanii* is reported to be an etiological agent of disease outbreaks in mandarin fish (*Siniperca chuatsi*), channel catfish (*Ictalurus punctatus*), Indian major carps (*Labeo rohita* and *Catla catla*) and Prussian carp (*Carassais auratus gibelio*). Other *Acinetobacter* species, such as *A. johnsonii, A. lwoffii, A. pittii, A. radioresistens*, and *A. junii*, were found to cause fish disease in recent years, suggesting that *Acinetobacter* spp. are emerging fish pathogens, posing a new threat to aquaculture [29,49].

The genus *Pseudomonas* includes more than 140 species, most of which are saprophytic. More than 25 species are associated with humans and in most cases cause opportunistic infections. These include *P. aeruginosa, P. fluorescens, P. putida, P. cepacia, P. stutzeri, P. maltophilia,* and *P. putrefaciens. P. aeruginosa* and *P. maltophilia* account for approximately 80 percent of pseudomonads recovered from clinical specimens. Because of the frequency with which it is involved in human disease, *P aeruginosa* has received the most attention. It is a ubiquitous free-living bacterium and is found in most moist environments. It infects a remarkably broad array of species, including plants, insects, and vertebrates. Although it rarely causes disease in healthy humans, it is a major threat to hospitalized patients, especially those with serious underlying conditions such as cancer, tuberculosis, AIDS, and severe burns. *P. aeruginosa* causes bacteremia and infection of the urinary tract, respiratory system, dermis, soft tissues, bones and joints, gastrointestinal tract, and blood [50,51,52]. According to the WHO, *P. aeruginosa* belongs to the most critical group of pathogens for which new antibiotics are urgently needed. *P. aeruginosa* is carbapenem-resistant [25]. *Pseudomonas fluorescens*, found in aquaponics by Chitmanat et al. [37], could be a rare cause of invasive hospital-acquired infections, with the usual site of infection being the bloodstream [53]. This bacterium, however, is commonly known for its strong food spoilage ability in ordinary and refrigerated food items (including milk and milk products, meat products, and vegetables) via its enzymes and pigment production [54]. Pseudomonads are one of the most dangerous fish pathogens that cause ulcerative syndrome and hemorrhagic septicemia. *P. aeruginosa* is part of the normal fish microbiota, but under stressful conditions this bacterium becomes highly opportunistic and pathogenic, causing serious disease [55]. On the other hand, *P. fluorescens* is found to cause chronic mortality in farmed Nile tilapia *(Oreochromis niloticus)* reared at low water temperatures [56]. Moreover, *P. aeruginosa, P. fluorescens, P. putida,* and *P. stutzeri* were among the common etiological agents of *Labeo rohita* and *Catla catla* infections [29]. In addition, *P. aeruginosa* is a typical plant pathogen capable of infecting the leaves and roots of the plant thale cress (*Arabidopsis thaliana)* [57].

The genus *Staphylococcus*, also found in aquaponics by Chitmanat et al. [37], consists of a number of species, of which *S. aureus, S. epidermidis,* and *S. saprophyticus* are the most frequently associated with human infection [58]. *S. aureus* is both a commensal bacterium and a human pathogen causing bacteremia and infective endocarditis as well as osteoarticular, skin and soft tissue, pleuropulmonary, and device-related infections [59]. According to the WHO, *S. aureus* belongs to the list of high priority pathogens for which new antibiotics are urgently needed. *S. aureus* is methicillin-resistant and vancomycin-intermediate-resistant [25]. Food handlers who have *S. aureus* on their skin and mucous membranes can act as a source of fish contamination during fish handling and processing [60]. On the other hand, *S. aureus* enterotoxins can cause gastroenteritis in humans via consumption of fish and fish products [13]. In addition, *S. aureus* is found to be pathogenic to *Arabidopsis thaliana* [57]. Recently, the opportunistic pathogen *S. epidermidis* has been the main cause of catheter-related bloodstream infections and early-onset neonatal sepsis as well as a frequent reason for prosthetic joint infections, prosthetic valve endocarditis, and other device-related infections [61]. This bacterium is reported to infect tilapia (*Oreochromis niloticus*), causing splenomegaly [56]. The widespread *S. saprophyticus* is both a commensal bacterium as well as a common human uropathogen associated with 10–20% of all urinary tract infections in sexually active women worldwide [62]. It was isolated from the liver and kidneys of freshwater-farmed hybrid sturgeon (*Acipenser baerii* × *Acipenser schrenckii*), and causes a disease with high mortality and surface bleeding [63].

*Micrococcus* spp. strains are commonly found in a wide variety of terrestrial and aquatic ecosystems, including soil, fresh and marine water, sand, and vegetation, as well as on the skin of warm-blooded animals, including humans. Also, *Micrococcus* spp. Strains have been reported to cause a variety of infections, usually as opportunistic pathogens. Thus, *M. luteus* strains are associated with septic arthritis, prosthetic valve endocarditis, and recurrent bacteremia. In addition, *Micrococcus* spp. Strains caused pneumonia in a patient with acute leukemia, localized skin infections in immunocompromised patients with HIV-1 disease, and catheter-related infection in patients with pulmonary arterial hypertension [64]. *M. luteus*, an important member of this genus, is found to cause disease in rainbow trout (*Oncorhynchus mykiss*), gilthead sea bream (*Sparus aurata*), sharpsnout sea bream (*Diplodus puntazzo*), and common dentex (*Dentex dentex*) [65]. This bacterium causes leafspot on mango as well [66].

*Clostridium* spp. were identified in an aquaponic system with multiple spoilage species. However, the species *C. botulinum* and *C. perfringens*, which cause foodborne disease outbreaks, were not detected by the authors [6]. Nevertheless, Khalil et al. [67] also found the presence of *Clostridium* in an aquaculture system, suggesting a potential risk of infection with pathogens of this genus in aquaponics. *C. perfringens* is widespread in nature; its primary habitat is the intestinal tract of humans and animals as well as the soil where feces are found [68]. Sabry et al. [69] found that in aquaculture there was a higher isolation level of *C. perfringens* from the external surface of fresh fish (31.8%) compared to the intestinal content of the same fish (9.1%). The virulence of *C. perfringens* results from the toxins produced by some of its strains. It causes several human diseases ranging from necrotizing enteritis to wound infection and life-threatening gas gangrene [69]. The other major pathogen of genus *Clostridium*, *C. botulinum*, is widely distributed in nature and occurs naturally in soil and aquatic environments. It is the cause for botulism due to the production of botulinum neurotoxin. Botulism is a severe human and animal disease characterized by flaccid paralysis leading to respiratory distress and death in the most severe cases. The presence of *C. botulinum* in fish can be associated with direct contact with contaminated aquatic environments and ingestion of *C. botulinum* spores from sediments or contaminated food. *C. botulinum* in fish can pose a significant threat to public health, especially when mishandling during fish processing occurs or insufficient heat treatment fails to destroy all *C. botulinum* spores in the final product [70,71]. Also, *Clostridium* spp. were associated with some soft rot diseases of herbaceous crops and sweet potato, as well as carrot and potato diseases [72].

Worthy of attention for the present work is the study of Khalil et al. [67] who examined the microbial population of two commercial recirculating aquaculture systems because they found multiple genera of bacteria and fungi not found so far in the studies of microbial communities of aquaponic systems. Since aquaculture systems are a part of aquaponic systems, the results obtained by these authors have been added to this review. They found the presence of genera *Acinetobacter, Bacteroides, Chryseobacterium, Legionella, Pseudomonas, Microbacterium, Clostridium*, and *Rhodococcus*, some of which were already mentioned above because they were discovered in aquaponics as well (Table 1).

*Bacteroides* spp. are important clinical pathogens which are found in most anaerobic infections in humans. They are the most predominant anaerobes in the human colon and of these *B. fragilis* is the most virulent [73]. Some *Bacteroides* species can play both beneficial and pathogenic roles based on their location in the human organism, often being beneficial in the gut but opportunistic pathogens elsewhere in the body where they cause bacteremia and abscess formation [73,74]. According to *Kabiri* et al. [75], the microbiome of fish fecal samples obtained from tilapia (*Oreochromis niloticus*), grass carp (*Ctenopharyngodon idella*), channel catfish (*Ictalurus punctatus*), and blue catfish (*Ictalurus furcatus*) included *B. eggerthii, B. uniformis, B. ovatus*, and *B. stercoris*. It should be noted that *B. ovatus* belongs to *B. fragilis* group and has pathogenic potential [76].

The species of *Chryseobacterium* are commonly found in environmental, food, and water sources and some have been isolated from the clinical environment, humans, and animals, while others are pathogenic for fish and humans. Some species such as *C. indologenes, C. soldanellicola, C. oranimense,* and *C. koreense* could be potential human pathogens, while *C. piscium* might be pathogenic to fish. *C. indologenes* is the most frequently isolated *Chryseobacterium* species from clinical specimens. It is a rare etiological agent of human disease, usually causing nosocomial infections that include bacteremia, pneumonia, meningitis, pyomyositis, keratitis, as well as indwelling device-related infections such as urinary tract, surgical, and burn wound infections. In foods, *Chryseobacterium* spp. are generally considered spoilage bacteria, as most are psychrotolerant and produce proteolytic enzymes, while some produce biogenic amines [77,78]. Also, *C. indologenes* is one of the primary causative agents of *Panax notoginseng* root rot [79].

*Legionella* spp. have been identified as one of the major causes of severe community-acquired pneumonia or nosocomial pneumonia. The most frequently isolated pathogenic species of this genus is the opportunistic pathogen *L. pneumophila*. The bathing facilities of public baths are the major source of infection [80,81]. *Legionella* infection occurs almost exclusively by aspiration of contaminated water while person-to-person transmission is very rare [82].

*Microbacterium* spp. are typically found in various environmental sources, such as soil and water samples. The most frequently isolated species from clinical samples are *M. oxydans* and *M. paraoxydans*. They rarely cause human infection, mostly infecting immunocompromised patients and catheter insertion sites, making them difficult to identify in clinical settings. *Microbacterium* spp. could be the cause for bacteremia, peritonitis, and endophthalmitis [83,84].

A *Rhodococcus* species with clinical importance is *R. equi*. Although it is mainly an opportunistic pathogen, a number of cases described infection occurring among individuals with normal immune systems. The main clinical symptom is pneumonia but this bacterium can disseminate to cause disease in virtually any human tissue [85]. Moreover, Speare et al. [86] reported a *Rhodococcus* sp. with pathogenic potential to juvenile Atlantic salmon *(Salmo salar*). In addition, *R. fascians* is a plant-pathogenic bacterium that causes malformations on aerial plant parts, whereby leafy galls occur at axillary meristems [87]. This species typically causes disease on herbaceous perennials [88].

In conclusion, the various human pathogenic bacteria found in aquaponics are usually of fish origin or end up in the water when proper hygiene practices are not applied and followed. It should be noted that only a few cases of *Salmonella* spp. and one incident of *E. coli* O157:H7 in aquaponics have been found in the literature so far. Moreover, no cases of *L. monocytogenes* have been reported in aquaponics until now. Most of the other pathogens found so far in aquaponics are opportunistic, and dangerous mainly to immunocompromised patients. However, many aquaponic pathogens are listed in the WHO list of drug-resistant bacteria for which new antibiotics are urgently needed. This could pose a challenge to infection treatment. Many of the human bacterial pathogens are pathogenic to fish but rarely to plants (Figure 1), which should be considered when preventive actions against disease are taken. In view of the aforementioned facts, if strict hygiene practices are followed, aquaponic systems are a good choice for growing healthy fish and plants safe for human consumption.

**Table 1 microorganisms-11-02824-t001:** Pathogenic bacteria in aquaponics that are potentially hazardous for human health.

Bacteria	Fish Species	Plant Type	Reference
**Gram-Negative**			
*Aeromonas hydrophila*	Hybrid catfish (*Clarias macrocephalus* × *C. gariepinus*); Nile tilapia (*Oreochromis niloticus*); Mozambique tilapia (*Oreochromis mossambicus*)	Lettuce (*Lactuca sativa*)	[37,89,90]
*Acinetobacter* spp.	Nile tilapia (*Oreochromis niloticus*)		[67]
*Acinetobacter baumanii*	hybrid catfish (*Clarias macrocephalus* × *C. gariepinus*)	Lettuce (*Lactuca sativa*)	[37]
*Bacteroides* spp.	Nile tilapia (*Oreochromis niloticus*)		[67]
*Chryseobacterium* spp.	Clarias (*Clarias gariepinus*); Nile tilapia (*Oreochromis niloticus*)		[67]
Coliforms	Rainbow trout (*Oncorhynchus mykiss*); lambari fish (*Astyanax bimaculatus*); tilapia (*Oreochromis niloticus* x *O. aureus*); Nile tilapia (*Oreochromis niloticus*); catfish *(Silurus glanis);* koi (*Cyprinus rubrofuscus*)	Duckweed (*Lemna minuta*); lettuce (*Lactuca sativa*)	[3,4,19,22,38,91,92]
*E. coli*	Rainbow trout*(Oncorhynchus mykiss*); hybrid catfish (*Clarias macrocephalus* × *C. gariepinus*); Nile tilapia (*Oreochromis niloticus*);	Duckweed (*Lemna minuta*); lettuce (*Lactuca sativa*)	[4,6,19,21,22,37,89,92]
*Legionella* spp.	Nile tilapia (*Oreochromis niloticus*)		[67]
*Pseudomonas* spp.	Nile tilapia (*Oreochromis niloticus*)		[67,89]
*Pseudomonas fluorescens*	Hybrid catfish (*Clarias macrocephalus* × *C. gariepinus*)	Lettuce (*Lactuca sativa*)	[37]
*Plesiomonas shigelloides*	Hybrid catfish (*Clarias macrocephalus* × *C. gariepinus*)	Lettuce (*Lactuca sativa*)	[37]
*Salmonella* spp.	Rainbow trout (*Oncorhynchus mykiss*); hybrid catfish (*Clarias macrocephalus* × *C. gariepinus*)	Duckweed (*Lemna minuta*); lettuce (*Lactuca sativa*)	[19,37]
*Shigella sonnei*	Mozambique tilapia (*Oreochromis mossambicus*)		[90]
**Gram-positive**			
*Clostridium* spp.	Nile tilapia (*Oreochromis niloticus*); Clarias (*Clarias gariepinus*)		[6,67]
*Microbacterium* spp.	Clarias (*Clarias gariepinus*)		[67]
*Micrococcus* spp.	Hybrid catfish (*Clarias macrocephalus* × *C. gariepinus*)	Lettuce (*Lactuca sativa*)	[37]
*Rhodococcus* spp.	Nile tilapia (*Oreochromis niloticus*)		[67]
*Staphylococcus* spp.	Hybrid catfish (*Clarias macrocephalus* × *C. gariepinus*)	Lettuce (*Lactuca sativa*)	[37]

### 2.2. Fungi and Fungus-Like Microorganisms (Oomycetes)

Filamentous fungi occur commonly in the environment due to their ability to grow on almost any substrate and under harsh conditions, and to produce spores that are dispersed in the air at low temperatures. They are heterotrophic and saprophytic organisms, extracting nourishment and energy from dead organic matter and possessing the ability to synthesize various natural products such as primary and secondary metabolites. These fungi can be dispersed in the environment in various ways, mostly by air, soil, water, and seeds, while the transmission route through insects acting as vectors is less frequent. *Aspergillus, Fusarium, Penicillium, Cladosporium, Acremonium, Alternaria*, and *Curvularia* are some of the most common fungal genera that belong to the filamentous fungi family, with the *Aspergillus* species reportedly most abundant and widespread worldwide [93,94]. Members of *Fusarium, Aspergillus*, and *Penicillium* genera, in particular, are known to produce secondary metabolites termed mycotoxins in specific conditions of temperature and humidity. The main mycotoxins produced by *Aspergillus* spp. are aflatoxins, and by *Fusarium* spp. are fumonisins, trichothecenes, and zearalenone. The main mycotoxin produced by *Penicillium* spp. is ochratoxin A. Consumption of mycotoxin-contaminated food or feed can lead to acute or chronic toxicity in humans and animals. Mycotoxins show genotoxic, carcinogenic, and mutagenic effects, and some of them have immunosuppressive activity [95,96]. Through aquaponic water, these mycotoxins produced by diseased plants could disseminate and accumulate in fish. Moreover, it has been confirmed that fungi can produce mycotoxins in water as well [97]. Finally, mycotoxins can enter the human organism through consumption of fish or/and plants produced by aquaponics.

*Aspergillus* spp. are filamentous fungi commonly found in soil, decaying vegetation, and seeds and grains (Table 2). Only a few well-known species of genus *Aspergillus* are considered important opportunistic pathogens in humans. *A. fumigatus* is the most common and life-threatening airborne opportunistic fungal pathogen which is particularly important for immunocompromised hosts. Inhalation of *A. fumigatus* spores (conidia) into the lungs can cause multiple diseases in humans that depend on the immunological status of the host. These diseases include invasive pulmonary aspergillosis, aspergilloma and various forms of hypersensitivity diseases such as allergic asthma, hypersensitivity, pneumonitis and allergic bronchopulmonary aspergillosis [98]. After *A. fumigatus, A. flavus* is the second leading etiological agent of invasive aspergillosis and it is the most common cause of superficial infection. The most common clinical syndromes associated with *A. flavus* include chronic granulomatous sinusitis, keratitis, cutaneous aspergillosis, wound infections, and osteomyelitis following trauma and inoculation. In addition, *A. flavus* produces aflatoxins, the most toxic and potent hepatocarcinogenic natural compounds ever characterized [99]. *A. niger* is a mold rarely reported as a cause of pneumonia [100]. Opportunistic plant infections by *Aspergillus* species are also common following drought, insect damage, or other environmental stresses. In particular, infection by *A. flavus* and *A. parasiticus* strains causes large economic losses in agriculture due to related contamination with mycotoxins. *A. flavus* is the main cause of *Aspergillus* infections and aflatoxin contamination of crops. In contrast, while *A. fumigatus* is the most common cause of human and veterinary aspergillosis, it is not known to cause disease in any host plant [101]. *A. niger* is a common phytopathogen that infects many fruits and vegetables, such as onion, corn, and others, causing destruction, rotting, and decomposition of plant tissues [102]. Moreover, *A. niger, A. flavus, A. ochraceus, A. terreus*, and *A. versicolor* were found to cause mycotic infections in freshwater Nile tilapia (*Oreochromis niloticus*) [103]. Therefore, it can be concluded that if *Aspergillus* spp. members break into an aquaponic system, serious disease to humans, plants, and/or fish is likely to occur.

*Fusarium* spp. show a global distribution and are associated with a wide range of emerging infections in plants, animals, and humans collectively termed fusariosis. In the medical field, various species of *Fusarium* are associated with local or invasive infections in both immunocompromised and immunocompetent individuals. The most prevalent infections are onychomycosis, skin infections, and keratitis. Among human pathogenic *Fusarium, F. solani* is the most common and virulent (comprising approximately 40–60% of infections), followed by *F. oxysporum* (~20%) and *F. fujikuroi* and *F. moniliforme* (~10%) [104,105]. Phytopathogenic *Fusarium* spp. are *F. oxysporum*, *F. solani*, *F. fujikuroi*, and *F. graminearum*. They infect a wide range of plants, including popular aquaponics plants such as tomato, cucumber, onion, spinach, pea, eggplant, and strawberry [106,107]. *F. moniliforme* and *F. udum* were found to be natural pathogens of freshwater fish reared in reservoirs, causing mycosis and high mortality [108]. In addition, the *F. solani* species complex causes superficial and systemic mycosis of Nile tilapia (*Oreochromis niloticus*) and zebrafish (*Danio rerio*) [109,110]. The abovementioned facts show that *Fusarium* spp. and especially *F. solani* are highly hazardous for aquaponics, due to their high pathogenic potential for plants, fish, and humans.

*Penicillium* spp. are multifarious and widespread in the environment but, despite their abundance and diversity, they are not often associated with human and animal infections. The species usually related to such infections are *P. citrinum*, *P. chrysogenum*, *P. digitatum*, *P. expansum*, and *P. marneffei* and the mode of infection is mostly via inhalation and sometimes ingestion. Diseases that result from *Penicillium* infection of any *Penicillium* species are commonly referred to as penicillosis. Species of this genus have been mentioned in relation to human infections such as keratitis, endophthalmitis, otomycosis, pneumonia, endocarditis, and urinary tract infections [93,111]. Additionally, *P. expansum* is a dominant post-harvest pathogen among fruits and vegetables [112] while *P. cyclopium*, *P. viridicatum*, *P. hirsutum*, and *P. allii* have been reported as garlic pathogens [113]. Some *Penicillium* spp. are highly hazardous for blue tilapia (*Oreochromis aureus*), Nile tilapia (*Oreochromis niloticus*), and Indian carp (*Catla catla*) [103,114,115]. In addition, Shahbazian et al. [116] isolated *P. expansum* and *P. citrinum* from infected eggs of rainbow trout *(Oncorhynchus mykiss*). These facts emphasize that the presence of *Penicillium* spp. in aquaponic systems results in a certain danger to human, plant, and fish health.

Unlike *Aspergillus*, *Fusarium*, and *Penicillium*, which cause severe infections in a wide variety of patients, *Trichoderma* has generally been considered nonpathogenic in humans, but localized and disseminated infections in immunocompromised and immunocompetent patients have been reported worldwide. *T. longibrachiatum* is the most frequently reported species associated with invasive fungal infections, followed by *T. atroviride*, *T. bissettii*, *T. citrinoviride*, *T. harzianum*, *T. koningii*, *T. pseudokoningii*, and *T. viride* [117]. *Trichoderma* spp. cause a variety of clinical manifestations, such as invasive pulmonary infection, peritonitis, CNS infection, endocarditis, fungemia, and disseminated disease affecting distant organs, particularly in patients with hematological malignancies and those undergoing long-term ambulatory peritoneal dialysis (CAPD) [117,118]. *Trichoderma* spp. are considered beneficial for aquaponically grown plants, because they are plant symbionts widely used as biofertilizers and biocontrol agents for plant diseases [119]. In addition, it has been reported that *Trichoderma* spp. extracts have antimicrobial activity against human and fish pathogens due to the secondary metabolites produced by these fungi [120]. Nevertheless, *T. asperellum* is considered a rare and low-pathogenic fungus in fish [121].

*Cladosporium* spp. are existing in both outdoor and indoor environments and they rarely cause illness in humans. Nevertheless, subcutaneous abscesses, central nervous system, and pulmonary infections in immunocompromised patients have been documented in the literature. One of the most frequently reported species with pathogenic potential is *C. bantiana*, followed by *C. sphaerospermum* [122,123]. *Cladosporium* spp. are associated with numerous agricultural crop diseases, causing leaf spots, scab, postharvest rots, and other symptoms leading to economic losses [124]. Also, they cause mycotic infections in freshwater Nile tilapia (*Oreochromis niloticus*) [103].

*Acremonium* spp. are commonly found in soil, rotting vegetation, and decaying food [125]. Species reported to cause human infections include *A. alabamensis*, *A. falciforme*, *A. kiliense*, *A. roseogriseum*, *A. strictum*, *A. potroni,* and *A. recifei*. This genus has been recognized as an etiological agent of human skin infections. Eumycotic mycetoma is caused by a variety of fungi, but not often by *Acremonium*. In addition, the most common pathogens of onychomycosis (fungal nail infection) are dermatophytes and *Fusarium* spp. followed by *Acremonium* spp. In the literature cases of keratitis, osteomyelitis, peritonitis and dialysis fistulae infection, localized infections, pneumonia, and disseminated infections, including meningitis, endocarditis, and cerebritis, have been published. Reports of systemic infections are almost always in patients with underlying risk factors such as malignancy and transplantation [125,126]. *A. strictum* is potentially pathogenic to the flower stems of weakened carrot plants under stress conditions [127].

The hyphomycetous genus *Phaeoacremonium* shows morphological characteristics between *Acremonium* and *Phialophora*. The main environmental source of these fungi are woody plants. The current list of human pathogens includes *P. alvesii*, *P. amstelodamense*, *P. griseorubrum*, *P. krajdenii*, *P. parasiticum*, *P. rubrigenum*, *P. sphinctrophorum*, *P. tardicrescens*, and *P. venezuelense*, of which *P. parasiticum* is the most frequently isolated from human hosts. Most reported cases of *Phaeoacremonium* infection included subcutaneous abscesses, cysts, or chronic or acute osteoarthritis in immunocompetent or immunocompromised patients; these cases were often initiated by traumatic inoculation. Disseminated infections, fungemia, or endocarditis have been found in a few cases involving immunocompromised patients [128,129].

The ubiquitous *Rhizopus* is the most common fungal genus causing mucormycosis; other less common etiological agents of infection include *Mucor* spp. and *Rhizomucor* spp. The infection usually affects immunocompromised patients and commonly presents two clinical syndromes: sinopulmonary and rhinocerebral. Some rare forms include cutaneous, intestinal, and pulmonary diseases [130]. The human pathogenic species of this genus include *R. microsporus* and *R. azygosporus* [131,132]. Additionally, it was reported that *R. arrhizus* caused rot on sunflower and tomato plants in China and Pakistan, respectively [133,134]. In addition, *Rhizopus* spp. are responsible for post-harvest fruit rot in strawberries in the UK [135]. *Rhizopus* spp. are one of the etiological agents of mycotic infections with economic importance in freshwater Nile tilapia (*Oreochromis niloticus*) [103].

The *Mucor* genus consists of early diverging fungi which are basal in comparison to higher fungi (i.e., Ascomycota and Basidiomycota phyla). *Mucor* species are very abundant in nature and often ubiquists. They could cause either superficial (cutaneous, subcutaneous) or invasive mycoses called mucormycoses, especially in immunocompromised patients. According to literature data, only three *Mucor* species are frequently cited in human infection cases: *M. circinelloides*, *M. indicus*, and *M. pusillus* [136]. *Mucor* is one of the genera responsible for postharvest rot of strawberry fruit [135,137]. Also, Reyes [138] reported a severe rot of tomato, cucumber, eggplant, and pepper infected by *M. mucedo. Mucor* spp. have been found to be highly pathogenic to cultured fish such as silver carp (*Hypophthalmichthys molitrix*) and goldfish (*Carassius auratus*) [139]. In addition, the aforementioned human pathogen *M. circinelloides* was found to be pathogenic to yellow catfish (*Pelteobagrus fulvidraco*) and zebrafish (*Danio rerio*) as well [140,141].

The *Microascus* genus comprises species commonly isolated from soil, decaying plant material and indoor environments. A few species are also recognized as phytopathogens and opportunistic pathogens in insects and animals, including humans [142,143]. *M. cirrosus* causes cutaneous infection [144], pulmonary infection [145], and fatal invasive infection with fungemia [146]. There are cases of human subcutaneous infection caused by *M. ennothomasiorum* [147], mycetoma caused by *M. gracilis* [148], and suppurative cutaneous granulomata caused by *M. cinereus* [149]. In general, *Microascus* spp. rarely cause infections and they usually occur in immunocompromised patients [146].

*Wallemia* spp. are known for their ability to grow in osmotically challenging environments, such as dry or salted foods, dry feed, indoor and outdoor air, etc. Up until now, only strains of *W. sebi*, *W. mellicola*, and *W. muriae* were related to human health problems as either allergological conditions or rare subcutaneous/cutaneous infections [150].

Genus *Macrophomina* was assigned to the Botryosphaeriaceae family and includes several phytopathogens: *M. phaseolina*, *M. pseudophaseolina*, *M. euphorbiicola*, and *M. vaccinia* [151]. The most important pathogenic species of this genus is *M. phaseolina*. This fungus rarely causes disease in humans. So far, there have been cases of keratitis and infections in a renal transplant recipient and a child with acute myeloid leukemia. Most of the cases were in immunocompromised patients [152]. *M. phaseolina* infects at least 500 plant species, causing diseases such as stem and root rot, charcoal rot, and seedling blight [153].

Yeasts exist in several environmental niches, including marine, aquatic, atmospheric, and terrestrial habitats. Yeasts in fish mucus and intestines are considered opportunistic pathogens that attack the fish organism when it is stressed or immunocompromised. Several genera of yeasts, isolated from fish and found in aquaponics, are considered human pathogens as well (e.g., *Candida*, *Cryptococcus*, *Debaryomyces*, *Rhodotorula*, and *Trichosporon* spp.) [154].

The black yeast-like fungi *Aureobasidium* are ubiquitous microorganisms found in a wide variety of environments as saprophytes, endophytes, and pathogens. They are known to be capable of producing numerous different metabolites, many of which find applications in the field of plant pathogen control [155]. The main pathogenic species of this genus are *A. pullulans* and *A. melanigenum*, which could be a source of infection in immunocompromised hosts. The reported clinical manifestations include keratomycosis, cutaneous mycoses, peritonitis, meningitis, and fungemia [156,157].

Yeasts of the genus *Candida* can be isolated from samples of groundwater, mineral water, domestic and industrial wastewater, rivers, and lakes, demonstrating the ubiquity of this genus and its ability to adapt to different environments. *Candida* species cause opportunistic infections in both healthy and immunocompromised individuals, including candidiasis, candiduria, and nail mycosis, as well as systemic infections that can debilitate patients or even lead to death [158]. Approximately 75% of all *Candida* infections in humans are caused by *C. albicans*, while *C. glabrata*, *C. parapsilosis*, and *C. tropicalis* are important emerging pathogens related to nosocomial infections. It is reported that *C. albicans* causes > 150 million mucosal infections and ~200,000 deaths per annum due to invasive and disseminated disease in susceptible individuals [158,159]. Furthermore, *C. albicans* is a cause of mycotic infections with economic importance in freshwater Nile tilapia (*Oreochromis niloticus*), common carp (*Cyprinus carpio*), catfish (*Clarias gariepinus*), and grey mullet (*Mugil cephalus*) [103,160]. In addition to *C. albicans*, other *Candida* species pathogenic to Nile tilapia (*Oreochromis niloticus*) include *C. parapsilosis* and *C. guilliermondii* [161]. Closely related to *Candida* spp. are *Debaryomyces* spp. with the most prominent member being *D. hansenii*. This yeast is commonly found in natural substrates and in many types of cheese. It has been repeatedly associated with catheter-related bloodstream infections and, rarely, with other infections [162]. *D. hansenii* demonstrated pathogenicity to goldfish (*Carassius auratus*), Atlantic salmon (*Salmo salar*), African catfish (*Clarias gariepinus*), and rainbow trout *(Oncorhynchus mykiss*) as well [154,163].

*Cryptococcus* spp. reside in diverse ecological niches. Both *C. neoformans* and *C. gattii*, the most important members of this genera, are abundant in decaying materials within hollows of different tree species. *C. neoformans* is particularly abundant in avian excreta. It is the main species of this genus and the predominant etiological agent of cryptococcosis, a globally distributed invasive fungal infection which presents substantial therapeutic challenges. The disease affects both immunocompromised and immunocompetent individuals and can cause pneumonia and meningoencephalitis [164]. *Cryptococcus* spp. has been found to be pathogenic to tilapia (*Oreochromis niloticus*) and African catfish (*Clarias gariepinus*) [154].

*Thichosporon* species are yeasts that are common in the environment and may be a part of the normal microbiota of the human skin and gastrointestinal tract. Additionally, they are often found to cause superficial infections of the skin, nails, and hair and invasive infection in immunocompromised patients. Also, a rare case of a *Trichosporon* brain abscess has been documented [165,166]. Among the most frequent etiological agents of *Trichosporon* infections are *T. inkin*, *T. asahii*, *T. cutaneum*, *T. mucoides*, *T. ovoides*, and *T. asteroides* [167]. In addition, *Trichosporon* spp. have been found to be pathogenic to tilapia (*Oreochromis niloticus*), gray mullet (*Mugil cephalus*), and African catfish (*Clarias gariepinus*) [154,161].

*Rhodotorula* are ubiquitous saprophytic yeasts that can be found in many environmental sources, as well as opportunistic pathogens that colonize and infect susceptible patients. Most of the cases of *Rhodotorula* infection in humans demonstrated symptoms of fungemia associated with central venous catheter use [168]. In addition, *Rhodotorula* spp. have been found to be pathogenic to tilapia (*Oreochromis niloticus*), African catfish (*Clarias gariepinus*), and gray mullet (*Mugil cephalus*) as well [154,161]. Closely related to *Rhodotorula*, *Sporobolomyces* are yeasts commonly isolated from environmental sources including lake water, tree leaves, and air with their natural habitat in humans, mammals, birds, and plants. *S. salmonicolor* is the most frequently isolated member of *Sporobolomyces* spp. from clinical samples, although *S. salmonicolor* infections are quite rare. This pathogenic yeast has previously been reported to cause invasive infections including dermatitis, cerebral infection, fungemia, encephalitis, ocular infection, and lymphadenitis [169]. As an opportunistic pathogen, *S. salmonicolor* can occasionally cause disease in fish as well [170].

Genus *Malassezia* comprises yeast species that are part of the normal human skin microbiota from where they can easily inhabit the environment, including aquaponic water. They are involved in skin disorders, such as pityriasis versicolor, seborrheic dermatitis, atopic eczema, and folliculitis [171].

*Sterigmatomyces* spp. are marine-derived yeasts belonging to the phylum Basidiomycota. Imashioya et al. [172] reported a rare case of liver abscess due to *S. halophilus* in a boy with acute lymphoblastic leukemia.

*Pythium* is a genus of fungus-like parasitic oomycetes. Although rare, human pythiosis caused by *Pythium insidiosum* has occurred in the USA, Thailand, Australia, New Zealand, Haiti, and Malaysia. The disease might be manifested as a vascular, ophthalmic, subcutaneous, or systemic type [173]. In addition, *P. insidiosum* is pathogenic to fish as well [174]. Moreover, *Pythium* spp. cause many plant diseases, including damping-off, root rot, collar rot, and stem rot in different production systems, including aquaponics and hydroponics [175,176].

In conclusion, most of the pathogenic molds found in aquaponics are dangerous to humans, plants, and fish (e.g., *Aspergillus*, *Fusarium*, *Mucor*, *Penicillium*, *Rhizopus*, *Cladosporium*, etc.). On the other hand, aquaponic yeasts are mainly pathogenic to fish and humans (Figure 1). Usually, the fungi found in aquaponics are described by the researchers in terms of genus, and in rare cases in terms of species. However, only some species of the fungal genus are pathogenic to humans. Thus, in most cases, the risk of the presence of human fungal pathogens in aquaponic water is not particularly high. From the analysis of different fungal pathogens, it can be concluded that they usually cause opportunistic infections in immunocompromised patients. As such, if proper hygiene practices are followed, the fungi in aquaponics are not a major concern for human health. In this regard, however, monitoring of mycotoxin content in fish and plants is necessary, since they can enter the human body through food consumption.

**Table 2 microorganisms-11-02824-t002:** Pathogenic fungi and oomycetes in aquaponics that are potentially hazardous for human health.

Fungi	Fish Species	Plant Type	Reference
**Molds**			
*Acremonium* spp.	Nile tilapia (*Oreochromis niloticus*)		[66]
*Aspergilllus* spp.	Clarias (Clarias gariepinus)		[66]
*Aspergilllus flavus*	Carp (*Cyprinus carpio*)		[177]
*Aspergilllus niger*	Carp (*Cyprinus carpio*)		[177]
*Cladosporium* spp.	Clarias (*Clarias gariepinus*)		[66]
*Fusarium* spp.	Clarias (*Clarias gariepinus*); Carp (*Cyprinus carpio*)	Lettuce (*Lactuca sativa*)	[66,175,177]
*Phaeoacremonium* spp.	Nile tilapia (*Oreochromis niloticus*)		[66]
*Macrophomina* spp.	Nile tilapia (*Oreochromis niloticus*)		[66]
*Microascus* spp.	Nile tilapia (*Oreochromis niloticus*)		[66]
*Mucor* spp.	Clarias (*Clarias gariepinus*)		[66]
*Penicillium* spp.	Nile tilapia (*Oreochromis niloticus*); Carp (*Cyprinus carpio*)		[66,177]
*Rhizopus* spp.	Carp (*Cyprinus carpio*)		[177]
*Trichoderma* spp.	Clarias (*Clarias gariepinus*); Carp (*Cyprinus carpio*)		[66,177]
*Wallemia* spp.	Clarias (*Clarias gariepinus*)		[66]
**Yeasts**			
*Aureobasidium* spp.	Nile tilapia (*Oreochromis niloticus*)		[66]
*Candida* spp.	Nile tilapia (*Oreochromis niloticus*)		[66]
*Candida albicans*	Carp (*Cyprinus carpio*)		[177]
*Candida parapsilosis*	Carp (*Cyprinus carpio*)		[177]
*Cryptococcus* spp.	Nile tilapia (*Oreochromis niloticus*)		[66]
*Debaryomyces* spp.	Clarias (*Clarias gariepinus*)		[66]
*Malassezia* spp.	Clarias (*Clarias gariepinus*)		[66]
*Rhodotorula* spp.	Clarias (*Clarias gariepinus*)		[66]
*Sterigmatomyces* spp.	Nile tilapia (*Oreochromis niloticus*)		[66]
*Sporobolomyces* spp.	Nile tilapia (*Oreochromis niloticus*)		[66]
*Trichosporon* spp.	Clarias (*Clarias gariepinus*); Nile tilapia (*Oreochromis niloticus*)		[66]
**Oomycetes**			
*Pythium* spp.		Lettuce (*Lactuca sativa*)	[175]

## 3. Microbial Pathogen Control in Aquaponic Systems

The microbiota of water is influenced by many factors, including the physicochemical properties of water, seasons, and climatic conditions [178]. In this regard, the origin of the water (whether it is rainwater, groundwater, drinking water, or originates from some other source) impacts microbial content and diversity [179]. There are a wide variety of human pathogenic bacteria, fungi, viruses, and oomycetes that can be spread through the water of aquaponic systems. In the literature are found six physical disinfection and filtration methods to mitigate pathogenic loads in aquaponics: ultraviolet irradiation (UV), blue light-emitting diodes (LED), media filtration, membrane filtration, heat, and sonication [180].

UV disinfection in soilless production systems involves exposing tank water to light in the germicidal range of roughly 225–312 nm. These lights usually disinfect effluent water from the fish tank but can be also used to disinfect influent water. The mechanisms of pathogen inactivation are damage to DNA and mRNA, with bacteria being the most susceptible to this damage. UV irradiation has adverse effects on exposed fish, in the form of skin lesions and reduction of goblet cells in the skin, which leads to reduced mucus production and downregulation of innate immunity [180,181,182].

Blue light-emitting (LED) photoinactivation involves placing these lights above fish tanks to expose the water. Blue light (400–500 nm) has a bactericidal effect and few if any harmful effects on the fish. Exposure of the fish pathogen *Edwardsiella piscicida* to 405 or 465 nm blue light for the specified exposure time was estimated to inactivate 99% of the bacteria [182].

Media filtration in soilless production systems involves pumping a nutrient solution through granular or fibrous material to capture and remove pathogens, with sand, rockwool or pozzolana as the most common filter bed materials. Such filters (pore size less than 10 µm) are known to separate various particles from the water, including microorganisms [180,183]. Media filters can either filter the incoming water before it reaches the production tanks or filter the effluent from production tanks to prevent recirculation of pathogens. Media composition, organic load, water temperature, and buildup of derris, as well as the rate of filtration (slow or rapid), determine the filter efficacy. Slow filtration is both a mechanical and biological process that uses water flow between 42 and 334 L/m^2^ h. Mechanical filtration occurs when particles larger than the pore size are prevented from moving through the filter media, while biological filtration takes place when the microorganisms in the water interact with those that grow on and within the media bed. The low water flow rates make the method of slow filtration insufficient for large soilless production systems. Nevertheless, slow filtration is the most frequently used filtration method because of its reliability and low cost. Microbial removal efficiency > 90% was usually achieved by this method [180,183,184].

Rapid media filtration processes handle water flow rates between 4167 and 19,792 L/m^2^ h [184]. However, in such a high flow rate debris builds up faster and the system requires backwashing more often. The basic limitations of rapid media filtration are the increased maintenance time and cost, compared to slow filtration. The benefit is the ability to handle the higher flow rates required for bigger soilless production operations [180,184].

Membrane filtration is another method of treating pathogen-laden water in aquaponic systems. Its principle is similar to media filtration. It involves forcing water through a woven or spun material that retains all substances larger than its pore size, with microfiltration membranes retaining particles >0.1 μm, ultrafiltration rejecting particles >0.01 μm, nanofiltration blocking particles >1 nm, and membrane reverse osmosis retaining materials >0.1 nm. The membrane is typically located upstream of the production tanks. Membrane filtration was shown to be effective against a variety of pathogens (fungi, bacteria, viruses, and nematodes) and the pore size of the membrane or the molecular weight cutoff is the key factor for filter efficacy [180,185].

Heat treatment is very effective against pathogens, achieving denaturation of their proteins resulting in the reduction of the initial microbial population by 90–99.9%. However, to suppress all kinds of pathogens, it is necessary to reach a temperature of 95 °C for at least 10 s. This practice consumes a lot of energy and requires water cooling (heat exchanger and transitional tank) before reintroduction of the treated water back into the irrigation loop. In addition, it has the major disadvantage of killing all microorganisms, including the beneficial ones [183].

In the context of soilless production systems, sonication includes application of high frequency waves (usually 20–40 kHz) to the influent nutrient solution as a means of inactivating pathogens prior to their entry into the production tanks. A machine is used to generate these waves, which are transmitted to the water via a probe. These waves induce the formation of low-pressure pockets inside the cells and cause them to collapse in a process called cavitation. The use of sonication is a relatively new method of disinfection and more research should be done on the overall efficacy of sonication as well as its applicability to soilless production systems [180].

Chemical treatments with ozone, hydrogen peroxide, and sodium hypochlorite are also administered in aquaponics. Ozonation has the advantage of eliminating all pathogens in certain conditions and of being rapidly decomposed to oxygen. However, ozone treatment is expensive and ozone irritates mucous membranes in the case of human or fish exposure. It also produces byproducts in raw water that should be removed, for example by UV radiation, prior to the return of the water to the fish tank [183]. Hydrogen peroxide (H_2_O_2_) is an oxidant that reacts forming water and oxygen radicals. Formic acid and acetic acid, which are commercially called activators, decrease pH in the nutrient solution to stimulate the reaction. This is an inexpensive but inefficient method useful for cleaning rather than disinfection. Nevertheless, doses of 0.01 to 0.005% are effective against *Pythium* spp., *Fusarium* spp, and other fungi [186,187]. Sodium hypochlorite (NaOCl) is an inexpensive and widely used chemical product for water treatment, especially in swimming pools [187]. When added to water, it reacts with the formation of Cl^–^ and O^+^ for strong oxidization of any organic material. Sodium hypochlorite is an effective agent against a number of pathogens such as *Fusarium* spp., but it is not effective against viruses [186].

From the above information it is clear that the proper choice of disinfectant, as well as its contact duration, correct and safe handling, and precise dosage should all be taken into consideration for the effective prevention of fish, plant, and human diseases. Also, conventional treatments need to be administered carefully because they could be deleterious to human, fish, plants, and beneficial microorganisms. Because of this, the ways to control the diseases are mainly based on preventive actions and physical water treatments. To limit the introduction of pathogens, actions such as cleaning and disinfection of the premises, specific clothes for the personnel, certified plant seeds, healthy fish, and established fish-handling techniques and diets, as well as physical barriers against insect vectors, are required. To limit and/or avoid the spread of pathogens, measures such as selection of resistant plant varieties, tool disinfection, avoidance of abiotic stresses, and good plant and fish densities should be introduced. Because the humid/aquatic environment suits almost every pathogenic fungus or bacterium, environmental management is required as well. In this respect, in large greenhouse structures such as aquaponics, computer software and algorithms are usually used to calculate the optimal parameters allowing for both plant and fish production and disease control. The parameters measured are, among others, temperature (air and water), humidity, vapor pressure deficit, wind speed, dew probability, leaf wetness, and ventilation. The practitioner acts on these parameters by manipulating heat, ventilation, shade, and light [6,13,183,186]. Nanoparticle-based sensors have also been adopted as tools for on-site environmental monitoring including online and real-time detection of microbial pathogens and other contaminants. For example, to detect *Xanthomonas axonopodis* pv. *vesicatoria*, which causes bacterial spotted disease in Solanaceae plants, fluorescent silica nanoparticles in combination with antibodies were developed [1]. If aquaponic system personnel become infected, timely medical consultation is very important, even if non-specific symptoms occur [13].

To date, there are no pesticides or biopesticides for plant pathogen control specifically developed for aquaponic use [183]. In this regard, however, antibiotics are frequently used in aquaculture for the prevention and control of disease and promotion of fish growth. Worldwide, the most commonly used therapeutic agents against aquaculture fish infections are oxytetracycline, chloramphenicol, and a sulfonamide-trimethoprim combination, as well as florfenicol and oxolinic acid, which are, however, also important for human medicine. They are mostly administered to fish through feed. Unconsumed medicated feed as well as fish excrement containing ingested antibiotics contribute to the leakage of these drugs into the rearing environment. Furthermore, prolonged antibiotic usage adds selective pressure to aquaponic microbiota and modulates water microbiomes. The resulting antibiotic resistance is a major problem for veterinary and human medicine [178,188]. In addition, the different antibiotics could cause leukopenia and thrombocytopenia, drug fever, allergy, neurological and pulmonary side effects, ventricular arrhythmia, nausea and vomiting, hepatitis, nephrotoxicity, and metabolic side effects [189].

Because of the deleterious effects of antibiotics, in last years they are being supplanted in aquaculture by other agents such as bacteriophages, which are applied directly to the tank water to eliminate pathogenic bacteria such as *Vibrio*, *Aeromonas*, *Pseudomonas*, and *Flavobacterium,* and to reduce fish mortality. This therapy has many distinctly beneficial features, such as high efficiency, specificity, and eco-friendliness, compared to antibiotics. Because all pathogenic bacteria form biofilms both in vitro and in vivo given the proper environmental conditions, bacteriophages could be applied to combat such biofilms, apart from infection treatment. Nevertheless, it is necessary to exercise caution with bacteriophage therapy since the possibility for bacteriophage resistance still exists. Currently, the functional use of bacteriophages against bacterial pathogens of cultured fish is still in its infancy and future research on the topic is necessary [180,190,191].

Another alternative to synthetic antimicrobial drugs in aquaculture and aquaponics is medicinal plants; their application to treat or prevent diseases is known as phytotherapy. It is a relatively simple and safe method for both the affected fish and the environment. The active compounds of plants are eco-friendly, biodegradable, and have few side effects on fish health. Phytotherapy is a medical practice that focuses more on traditional approaches rather than on modern medication. For example, the garlic plant (*Allium sativum*) can be used in the form of oral administration or medicinal baths to treat fish diseases caused by *Aeromonas hydrophila*, *Gyrodactylus tumbulli,* and *Neobenedenia* spp. [192].

It is known that as the microflora of an aquaponic system develops, it creates an environment that can suppress microbial pathogens as a result of natural antibiotic compounds released by beneficial microorganisms named probiotics. As such, biological control with antagonistic microorganisms against pathogens is a promising alternative to the aforementioned chemical and physical methods. Such biological control agents in aquaculture include strains of *Pseudomonas* and *Bacillus* species [23,183,186,193,194]. Applied directly to the tank water or with the fish feed, probiotics can control digestive problems and infections by preventing the proliferation of pathogenic bacteria from the fish gut. In addition, they can improve the digestion and absorption of nutrients in fish, resulting in improved growth rates and feed conversion efficiency and reduced risk of disease and mortality. Supplementation of probiotics to aquaponic systems not only results in enhanced fish growth but is beneficial to the plants as well [180,195]. After thorough analysis of the literature data, Yousuf et al. [196] found probiotics to be a viable alternative to antibiotics and other chemotherapeutics for the treatment of aquaculture fish infections, and one which could replace them to a great extent. In this respect, the future belongs to “Omics” technologies such as metagenomics and metatranscriptomics analysis that could elucidate the structure, metabolic functions, and interactions of microbial communities for a better identification of microbial strains and their metabolites with specific probiotic properties, thus providing effective biological control agents [186].

Another area for future exploration is the application of nano drugs for the treatment or prevention of infectious diseases in aquaponics. Because of their small size, nanoparticles are inhaled and cross brain membranes. The nano drugs are administered at lower levels, with remarkable improvement in their pharmacological effects, including enhanced bioavailability, more effective absorption, and reduced adverse effects. Nevertheless, the high price tag is a major limitation for nanomedicine application, as well as the concerns for short- and long-run influence on the body and the environment, which necessitates further studies [197].

## 4. Conclusions

The presence in the aquaponic system of numerous pathogens specific only to humans indicates that one of the most important ways of introducing such pathogens in aquaponics is the non-compliance of personnel with the established hygiene practices. It is also indispensable to choose healthy plants and fish, and for the water to meet the necessary requirements of aquaponics in terms of purity, composition, and microbial content. Some studies showed the potential of internalization of dangerous pathogens through the roots of hydroponically grown plants. However, the plants grown in soil showed more internalization events than those grown hydroponically. Nevertheless, more research in this regard is required to draw conclusions. It should be noted that many of the aquaponic pathogens are listed in the WHO list of drug-resistant bacteria for which new antibiotics are urgently needed. In addition, pathogen control by conventional physical methods, chemical methods, and antibiotic treatment could be hazardous to humans, fish, plants, and beneficial microorganisms, making biological control with antagonistic microorganisms, phytotherapy, bacteriophage therapy, and nanomedicine a promising alternative of these methods. However, information on these topics is rather scarce, so future investigations in this direction are necessary.

## Figures and Tables

**Figure 1 microorganisms-11-02824-f001:**
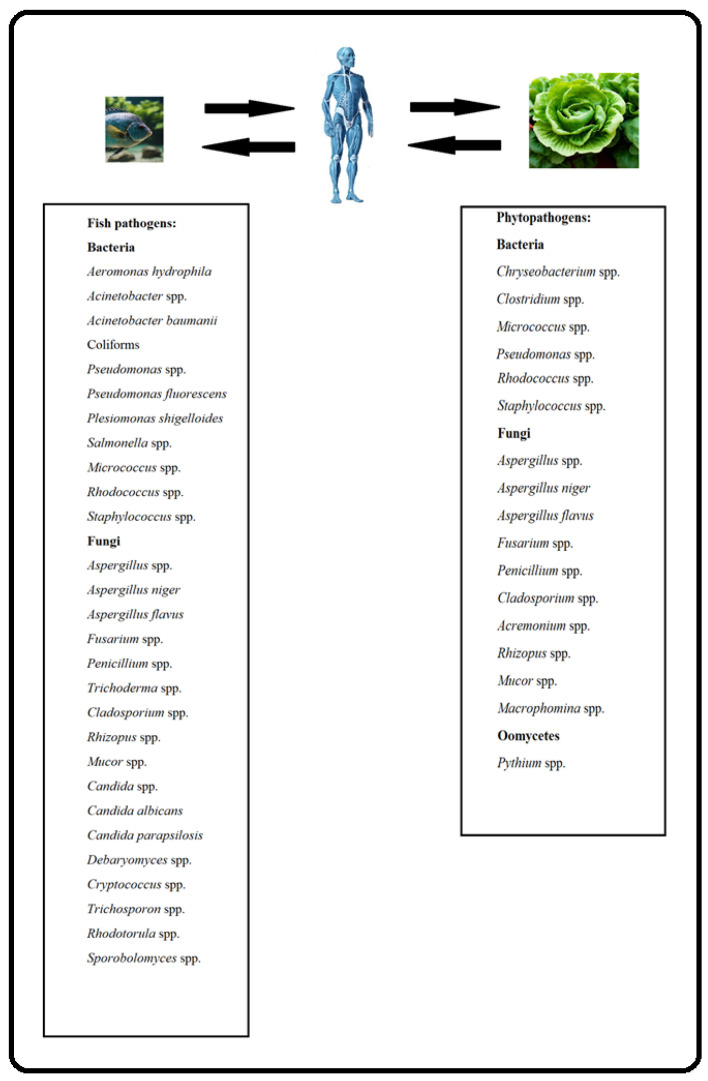
Fish and plant pathogens in aquaponics that are potentially hazardous for human health.

## Data Availability

All data is contained within the article.

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
