# Peer review of "Microbial Pathogens in Aquaponics Potentially Hazardous for Human Health"

_microorganisms, 2023, doi:10.3390/microorganisms11122824_

Round 1
Reviewer 1 Report
Comments and Suggestions for Authors
Line 78: 20 003 or 20003?
Line 105: put names of E. coli and Salmonella in italic, as well as in line 114 (Salmonella) and line 226 (S. aureus)
Line 157: separate abovementioned to above mentioned
In part 3. Microbial Pathogens in Aquaponic Systems I suggest that you add the importance of the influence and origin of the water on the presence of some microorganisms, in terms of whether water from a well, public supply, rainwater or some other water is used. Also, about the influence of ambient temperature and humidity and its control in the aquaponic system on the presence of fungi in plants.
Author Response
First of all, we should thank you for your remarks, which resulted in improved quality of the manuscript.
Line 78: 20 003 or 20003? – It is corrected in the text.
Line 105: put names of E. coli and Salmonella in italic, as well as in line 114 (Salmonella) and line 226 (S. aureus) – It is corrected in the text.
Line 157: separate abovementioned to above mentioned – It is corrected in the text.
In part 3. Microbial Pathogens in Aquaponic Systems I suggest that you add the importance of the influence and origin of the water on the presence of some microorganisms, in terms of whether water from a well, public supply, rainwater or some other water is used. – It is added in the text (Lines 664-666).
Also, about the influence of ambient temperature and humidity and its control in the aquaponic system on the presence of fungi in plants – It is added in the text (Lines 756-763).
Reviewer 2 Report
Comments and Suggestions for Authors
The manuscript addresses a highly interesting and current theme that encompasses the state of the art regarding contaminating microorganisms in aquaponic systems, and characterizes the diseases caused by them in different species (fish, animals, and humans). I believe that this theme is of great interest to the international scientific community, as fisherie consumption should be encouraged, as well as its production, to meet the hunger combat guidelines set by FAO/ONU, while ensuring no risks to human health.
Although I appreciated the evaluated literature review, its reading is somewhat tiresome and repetitive in terms of its structure. In sections 2 and 3, the same line of reasoning and phrases are presented for each of the species covered, making the reading less engaging. I believe the authors could have put more effort into ensuring a fluent and enjoyable reading experience for the audience. However, it is not incorrect in the way it was developed.
However, one of the most interesting topics (topic 3) was poorly addressed. It was written succinctly and did not fulfill the objective of providing a comprehensive background on how to prevent possible microorganisms. There is only a generic and shallow approach. Perhaps, by being more concise in topic 2, they could have provided a much more suitable background on the topic covered in topic 3. Topic 3 MUST be worked on more thoroughly.
Also, regarding topic 3, the authors mention the use of antibiotics, but did not provide any information or discussion about the occurrence of resistance in the microorganisms detected in this production system. Including this theme would make the work more relevant and with greater citation potential. I suggest its inclusion.
In the introduction section, the authors should provide information about the status of aquaponics worldwide. Is it widely used? In which areas? What are the main species produced? What is the volume of vegetables and aquatic organisms produced? Are the products already commercially available?
For me, it is also not clear whether aquaponics is used solely for fish production, or other aquatic organisms, and this needs to be made very clear in the manuscript.
My minor remarks are described below:
• The title seems inappropriate to me. Does the manuscript specifically address pathogens or microorganisms with potential risk to human health? It seems redundant and needs to be modified.
• In the Keywords section, do not repeat words already in the title for better article indexing.
• L76-81 – Please also include information about illnesses transmitted by the consumption of fish and other aquatic animals.
• L92 – Tilapia – Include the scientific name for all species mentioned in the manuscript. The same occurs in line 122, for example.
• L95, 105 –Salmonella, E. coli - use italics. Please double-check the use of italics in all scientific names.
• L112-114 – Describe which subspecies are relevant.
• L122 – Do not use "etc" in a literature review. Instead, include all available information.
• L137-149 – I believe that since bovines are the main reservoirs of EHEC, contamination in hydroponic systems is rarer compared to soil. Most vegetables involved in outbreaks in the United States contaminated by EHEC were grown near confinement areas. This may help explain these differences and should be mentioned in the review. Can you find information about this topic?
• L160-172 – The information is important, but it is not related to aquaponics at any point. This happens in other paragraphs of the review. Try to highlight the relevance to the theme of the review.
• L634-637 – No information was provided to support this statement.
Author Response
First of all, we would like to thank you for your in-depth review, which resulted in greatly improved quality of the manuscript.
The manuscript addresses a highly interesting and current theme that encompasses the state of the art regarding contaminating microorganisms in aquaponic systems, and characterizes the diseases caused by them in different species (fish, animals, and humans). I believe that this theme is of great interest to the international scientific community, as fisherie consumption should be encouraged, as well as its production, to meet the hunger combat guidelines set by FAO/ONU, while ensuring no risks to human health.
Although I appreciated the evaluated literature review, its reading is somewhat tiresome and repetitive in terms of its structure. In sections 2 and 3, the same line of reasoning and phrases are presented for each of the species covered, making the reading less engaging. I believe the authors could have put more effort into ensuring a fluent and enjoyable reading experience for the audience. However, it is not incorrect in the way it was developed.
However, one of the most interesting topics (topic 3) was poorly addressed. It was written succinctly and did not fulfill the objective of providing a comprehensive background on how to prevent possible microorganisms. There is only a generic and shallow approach. Perhaps, by being more concise in topic 2, they could have provided a much more suitable background on the topic covered in topic 3. Topic 3 MUST be worked on more thoroughly. – Topic 3 is much more thoroughly described (Lines 664-666; 671-745; 752-776; 779-805; 810-832).
Also, regarding topic 3, the authors mention the use of antibiotics, but did not provide any information or discussion about the occurrence of resistance in the microorganisms detected in this production system. Including this theme would make the work more relevant and with greater citation potential. I suggest its inclusion. - It is added in the text (Lines 24-26; 151-153; 158-162; 175-180; 216-219; 226-228; 237-239; 259-261; 279-282; 385-387; 843-845).
In the introduction section, the authors should provide information about the status of aquaponics worldwide. Is it widely used? In which areas? What are the main species produced? What is the volume of vegetables and aquatic organisms produced? Are the products already commercially available? - It is added in the text (Lines 40-41; 58-71 ).
For me, it is also not clear whether aquaponics is used solely for fish production, or other aquatic organisms, and this needs to be made very clear in the manuscript. - It is added in the text (Lines 66-67).
My minor remarks are described below:
- The title seems inappropriate to me. Does the manuscript specifically address pathogens or microorganisms with potential risk to human health? It seems redundant and needs to be modified. – The title is specified in this way because the term pathogens include not only microbial pathogens but protozoa and parasites as well. And the topic of this review are the microbial pathogens.
- In the Keywords section, do not repeat words already in the title for better article indexing – the problem is that the title already includes all important keywords. – New keywords are added (Line 33).
- L76-81 – Please also include information about illnesses transmitted by the consumption of fish and other aquatic animals. – Such information is added (Lines 96-101).
- L92 – Tilapia – Include the scientific name for all species mentioned in the manuscript. The same occurs in line 122, for example. – This information is added to the text.
- L95, 105 –Salmonella, E. coli - use italics. Please double-check the use of italics in all scientific names. – It is corrected in the text.
- L112-114 – Describe which subspecies are relevant – Unfortunately, these species were not listed in the article we cited.
- L122 – Do not use "etc" in a literature review. Instead, include all available information. It is corrected in the text. - It is corrected in the text.
- L137-149 – I believe that since bovines are the main reservoirs of EHEC, contamination in hydroponic systems is rarer compared to soil. Most vegetables involved in outbreaks in the United States contaminated by EHEC were grown near confinement areas. This may help explain these differences and should be mentioned in the review. Can you find information about this topic? – It’s true that enterohaemorrhagic E. coli circulate in cattle farms (such literature data is available). Therefore, in real life, conventional agriculture farms should have higher probability to meet and internalize this pathogen during the plant production and therefore in these farms would exist higher probability of infection of the consumer compared to hydroponics and aquaponics. However, in this trial of Macarisin et al. (2014), the soil was intentionally contaminated by E. coli O157:H7 not just randomly taken from the field. Also, the pathogen was introduced in the hydroponic system. So, the circulation of bovine EHEC in nature could not explain the aforementioned experimental result.
- L160-172 – The information is important, but it is not related to aquaponics at any point. This happens in other paragraphs of the review. Try to highlight the relevance to the theme of the review. – The purpose of these 2 paragraphs is to elaborate on Plesiomonas spp. and Shigella sonnei, which are not typical for aquaponics but are found there (Table 1), probably through human (or insect) vectors. There are other such cases as well, which highlight the necessity to follow the established hygiene practices. We included such cases in this review because these pathogens are also human health hazards.
- L634-637 – No information was provided to support this statement. – Such information is written in lines 165-169 and 746-763. Also, the presence in aquaponics of typical human pathogens support this statement.
Reviewer 3 Report
Comments and Suggestions for Authors
Please add solid conclusion in abstract.
There are also other Gram negative bacterial pathogens, causing diseases in humans
For examples: Edwardsiella tarda https://pubmed.ncbi.nlm.nih.gov/18804863/
Yersinia ruckeri https://pubmed.ncbi.nlm.nih.gov/26404907/
Add legends in tables
The authors did not discuss in detail about what kind of phyto-therapeutic compounds can use for the treatment of pathogens.
Any idea about nano medicine for controlling the pathogens.
Please add in tables, what kind of human diseases are caused by fish pathogens.
Add some human disease pictures caused by fish pathogens.
Please make a workflow to understand your concept.
In conclusion, please provide more information or new technologies to control pathogens.
Comments on the Quality of English LanguageMinor editing of English language required
Author Response
First of all, thank you for your remarks, which we think resulted in improved quality of the manuscript. We made the corrections to the best of our ability,
Please add solid conclusion in abstract. It is added in the text (Lines 24-30).
There are also other Gram negative bacterial pathogens, causing diseases in humans
For examples: Edwardsiella tarda https://pubmed.ncbi.nlm.nih.gov/18804863/
Yersinia ruckeri https://pubmed.ncbi.nlm.nih.gov/26404907/ - Indeed, there are some Gram-negative bacterial pathogens of fish or plant origin causing diseases in humans that are not covered in this review article, as the purpose of this review is to describe pathogens found so far in aquaponics (as stated in the title), not just fish and plant pathogens. Also, some of the pathogens found in aquaponics are not of fish or plant origin, but typical human pathogens, indicating the role of hygiene practices in aquaponics for the prevention of human disease.
Add legends in tables
The authors did not discuss in detail about what kind of phyto-therapeutic compounds can use for the treatment of pathogens. – This information is added to the article (Lines 797-805).
Any idea about nano medicine for controlling the pathogens. - This information is added to the article (Lines 763-768; 825-832).
Please add in tables, what kind of human diseases are caused by fish pathogens – this information is incorporated into and elaborated upon in the text. To avoid repetition, we would not like to add additional tables.
Add some human disease pictures caused by fish pathogens – It is possible to add such pictures but then the question is for which diseases? There are many human diseases caused by fish pathogens, which we described in this article, so instead we decided to prepare a figure which includes all pathogens of fish or plant origin potentially hazardous to human health, to summarize and illustrate this quite diverse information (Lines 552-553).
Please make a workflow to understand your concept.
In conclusion, please provide more information or new technologies to control pathogens. –This information is added to the text (Lines 785-832).